# Prolonged Survival after Recurrence in HCC Resected Patients Using Repeated Curative Therapies: Never Give Up!

**DOI:** 10.3390/cancers15010232

**Published:** 2022-12-30

**Authors:** Cyprien Toubert, Boris Guiu, Bader Al Taweel, Eric Assenat, Fabrizio Panaro, François-Regis Souche, Jose Ursic-Bedoya, Francis Navarro, Astrid Herrero

**Affiliations:** 1Department of HBP Surgery and Liver Transplantation, Montpellier University Hospital, University of Montpellier, 34295 Montpellier, France; 2Department of Digestive Imaging, Montpellier University Hospital, University of Montpellier, 34295 Montpellier, France; 3Department of Digestive Oncology, Montpellier University Hospital, University of Montpellier, 34295 Montpellier, France; 4Department of Minimally Invasive and Oncologic Surgery, Montpellier University Hospital, University of Montpellier, 34295 Montpellier, France; 5Liver Transplantation Unit, Department of Hepatology, Montpellier University Hospital, University of Montpellier, 34295 Montpellier, France

**Keywords:** HCC recurrence, survival after recurrence, treatment at recurrence, liver resection, liver transplantation, thermal ablation, microvascular invasion

## Abstract

**Simple Summary:**

Surgical resection is the optimal treatment for hepatocellular carcinoma (HCC), despite a high risk of recurrence. We performed a retrospective analysis of survival after recurrence (SAR) of HCC after liver resection. Time to recurrence and treatment at recurrence are major prognostic factors of SAR. However, after curative treatment at recurrence, Overall survival is not significantly different between patients who recurred or not, whatever the time of recurrence.

**Abstract:**

Surgical resection is the optimal treatment for HCC, despite a high risk of recurrence. Few data are available on patient’s survival after resection. This is a retrospective study of tumor recurrence occurring after hepatectomy for HCC from 2000 to 2016. Univariate and multivariate analyses were performed to identify prognostic factors of survival after recurrence (SAR). Among 387 patients, 226 recurred (58.4%) with a median SAR of 26 months. Curative treatments (liver transplantation, repeat hepatectomy, thermal ablation) were performed for 44.7% of patients. Independent prognostic factors for SAR were micro-vascular invasion on the primary surgical specimen, size of the initial tumor >5 cm, preoperative AFP, albumin and platelet levels, male gender, number, size and localization of tumors at recurrence, time to recurrence, Child–Pugh score and treatment at recurrence. In subgroup analysis, early recurrence (46%) was associated with a decrease in SAR, by contrast with late recurrence. However, the overall survival (OS) of patients with early recurrence and curative treatment did not significantly differ from that of non-recurring patients. For late recurrence, OS did not significantly differ from that of non-recurring patients, regardless of the proposed treatment. Aggressive and repeat treatments are therefore key to improve prognosis of patients with HCC.

## 1. Introduction

Liver cancers represent a major public health problem with an increase in incidence and associated mortality estimated at 1 million in 2030 and 1.5 million in 2060. They represent the 6th leading cause of cancer worldwide and the 3rd leading cause of cancer mortality in 2020, with 905,700 new cases worldwide in 2020, and are responsible for 830,200 deaths. Hepatocellular carcinoma (HCC) is the main histological type of primary liver tumor (75–85%) [1,2].

The optimal treatment for HCC on cirrhotic liver remains liver transplantation with five-year survival rates close to 70%, by respecting the selection criteria [3,4]. Nonetheless, because of organ shortages, there is a long waiting time carrying a high risk of dropout for tumor progression (8–24%) [5]. Surgical resection is the curative treatment option of choice for single lesions, regardless of size, on non-cirrhotic or cirrhotic liver with preserved liver function, without portal hypertension and a good general condition. Management of HCC on cirrhosis is currently guided by the Barcelona Clinic Liver Cancer (BCLC) classification [6], used in current practice and recommended by the American Association for the Study of Liver Diseases (AASLD) [7] and the European Association for the Study of the Liver (EASL) [8]. It differentiates the very early (0), early (A), intermediate (B), advanced (C) and terminal (D) stages of the disease, taking into account the tumor oncological criteria, the stage of cirrhosis and the general condition of the patient.

Despite the progress in liver surgery, imaging and oncology, the main problem related to the surgical treatment by resection for HCC is recurrence. Indeed, the recurrence rate at five years in patients operated on HCC for curative purposes is around 50–60% in the literature. Moreover, recurrence pejoratively influences overall survival after liver resection for HCC [9,10,11,12].

Most of the previous studies identify the presence of vascular invasion (both macro-vascular and micro-vascular), satellite nodules, large tumor size, high AFP level and its progression [13,14], multiple tumors, poor cell differentiation, advanced pathological tumor-node-metastasis (pTNM) stage [15,16,17] and some histological subtypes, such as macrotrabecular-massive HCC [18,19], as the main risk factors of recurrence.

The prognosis of HCC patients is associated with two intertwined variables: recurrence and underlying liver pathology until cirrhosis.

Several studies analyzed survival after recurrence (SAR) in this context [9,20,21] and highlighted the importance of a curative treatment of recurrence allowing very satisfactory overall survival rates, regarding those in patients without recurrence. Recurrence would therefore have no impact on survival if it can be managed curatively, either by liver transplantation [22], or by repeated liver resection [15,23] or by thermal ablation [16,24].

In our study, we also decided to analyze recurrence after liver resection in a large single-center cohort of patients over a period of more than 15 years, using SAR as the major endpoint, because it is less influenced by cirrhosis-related mortality than overall survival from initial surgery [25]. Furthermore, we analyzed the recurrence patterns and survival according to the type of treatments stratified on the time to recurrence.

## 2. Materials and Methods

We conducted a retrospective single-center cohort study on patients who were treated by first-line liver resection or liver resection while waiting for liver transplantation on cirrhotic or non-cirrhotic liver between 1 January 2000 and 31 December 2016.

The positive diagnosis of preoperative HCC was reached through the use of a pre-operative biopsy or according to EASL imaging criteria [8]. The pathological analysis of the surgical specimen subsequently confirmed this diagnosis.

Excluded from the study were patients < 18 years, patients with history of prior HCC and patients with other tumor histologies on the surgical specimen (such as hepato-cholangiocarcinoma, hepatic metastasis, benign tumor).

Our study was approved by our Institutional Review Board (IRB number: 202000515, date of approval: 16 July 2022).

### 2.1. Recurrence

After surgery, the patients underwent follow-up, which consisted of clinical examination, cross-sectional contrast-enhanced imaging (CT or MRI) every 3 months for the first 2 years, then every 6 months, associated with a chest CT. Serum alpha-fetoprotein (AFP) levels were obtained at the time points. The diagnosis of recurrence was made according to the same modalities as the initial diagnosis, according to the BCLC algorithm.

When recurrence was confirmed, it was classified as “early recurrence” or “late recurrence”, with a cut-off calculated according to the “minimum *p*-value” method. This method, which has been used in other oncological studies on HCC or other cancers [26,27,28], provides the best threshold expressed in months, based on the largest most-significant difference in survival after recurrence (SAR) between early and late recurrence after log-rank test, thereby minimizing ranking bias.

The choice of treatment for recurrence was discussed in a multidisciplinary consultation meeting, according to the BCLC algorithm for the management of HCC [6]. A curative treatment was offered whenever possible, depending on the characteristics of the recurrence (size, number and location of nodules), the general condition of the patients, their liver function and portal hypertension. For early HCC (BCLC 0-A), thermal ablation was the proposed first-line treatment, except when technical considerations (tumor location, visibility, accessibility) or prior history of endoscopic retrograde cholangiopancreatography (ERCP) dictated otherwise.

### 2.2. Clinical Variables

The data collected comprised: (i) preoperative demographic descriptive variables with age, gender, body mass index (BMI), diagnostic circumstances, American Society of Anesthesiologists (ASA) score, underlying liver disease and etiology, degree of associated liver fibrosis according to the METAVIR classification [29], cirrhosis, comorbidities, biological data characterizing biological liver function, expressed using Child–Pugh and albumin-bilirubin (ALBI) scores [30] and portal hypertension defined as the presence of splenomegaly with esophageal varicosities; (ii) tumor-related factors including tumor number, tumor size, preoperative alpha-fetoprotein (AFP) level, BCLC classification and AFP score [4]; (iii) the operative data collected including the carrying out of a preoperative treatment, the date of surgery, the extent of hepatectomy (hepatectomy consisting of the removal of three or more segments was deemed a major hepatectomy), the anatomic or not anatomic resection the surgical approach (laparoscopy or laparotomy), the duration of the operation, blood loss and perioperative transfusion; (iv) post-operative data including the collection of complications, the calculation of the “Comprehensive Complication Index” (CCI Score) [31] (a score ≥ 26.2 is the equivalent of at least one complication classified IIIa as a severe complication in the CLAVIEN-DINDO classification) and the length of stay in intensive care and hospital; and (v) histological data of the surgical specimen including size and number of lesions, micro and macrovascular invasion, satellite nodules, tumor capsule and invasion of adjacent organs. The resection was classified R1 if the resection margin was <1 mm.

The same data were collected for recurrence at the time of diagnosis. The localization of the recurrence made it possible to classify them into intrahepatic (IH), extrahepatic (EH) or mixed (M) recurrence. Intrahepatic recurrences were classified into 4 types according to their location with respect to the initial tumor (1: recurrence in contact with the parenchymal section, 2: recurrence in an adjacent segment, 3: recurrence in a distant segment, 4: multifocal recurrence). This classification was described by Poon et al. [32].

Modalities for the treatment of recurrence were also collected, including liver transplantation, repeat hepatectomy, thermal ablation, chemoembolization (TACE), systemic therapy and supportive care. Liver transplantation, repeat hepatectomy and thermal ablation have been defined as curative treatments, while chemoembolization and systemic treatments have been deemed non-curative treatments.

For survival data, we collected the dates of recurrence and death, or the latest news of the living patients at the end of the study.

### 2.3. Endpoints

The main objective was the analysis of prognostic factors of survival after recurrence (SAR). Secondary objectives were both a detailed analysis of the treatments for recurrence and of their impact on SAR, and a subgroup analysis by time to recurrence.

### 2.4. Statistical Analysis

Survival analyses are performed using Cox models for multivariate analyses, the Kaplan–Meier model and the log-rank test for comparisons in univariate analyses with a degree of significance at *p* < 0.05. Prognostic factors with a *p* < 0.10 in univariate analysis were included in a multivariate Cox model with a degree of significance at *p* < 0.05.

Subgroup comparative analyses were performed using the KHI 2 or the Fischer tests. Comparative analyses of multivariate subgroups were conducted using a logistic regression model. For comparisons of quantitative data, the Student’s test was applied.

The statistical analyses were conducted using IBM’s SPSS V.26 software (IBM Corp, Armonk, NY, USA) and survival curves using GraphPad Prism 8.4.3 for Window (GraphPad Software, San Diego, CA, USA).

## 3. Results

Between 1 January 2000 and 31 December 2016, 387 patients, who underwent curative partial liver resection for hepatocellular carcinoma (HCC), were analyzed. The mean age was 63.8 years (standard deviation (SD): 10.23), with a sex ratio (M/F) of 4.8.

The 90-day mortality rate was 5.4%. HCC was developed in liver cirrhosis in 258 (66.6%) patients. Resection was anatomical with the removal of the entire portal territory in 254 cases (65.9%).

Major hepatectomy was performed in 124 (32%) patients. They consisted of 85 right hepatectomies, 15 left hepatectomies and 24 trisectionectomies.

As for the minor resections performed on 263 patients (68%), 138 were anatomical and consisted of 80 bisegmentectomies (29 left lateral sectionectomies) and 58 mono-segmentectomies. The remaining 125 minor resections were non-anatomical: they included 80 and 45 wedges resections of 1 and 2 segments, respectively.

After a median follow-up of 46 months, 226 (58.4%) patients recurred (the rate of loss of follow-up was 3.9%).

In the entire cohort, HCC was single in 307 (79.3%) cases, the mean tumor size was 39.95 mm (SD: 34.09).

The demographic characteristics, management, operative data and the histological analysis of surgical specimens are illustrated in Table 1 for the general population and for the patients with or without recurrence.

### 3.1. Overall Survival and Recurrence-Free Survival in the Entire Cohort

The median overall survival (OS) of the cohort was 49 months (standard error 4.18 95% confidence interval [40.80–57.20]). The 1-, 3- and 5-year OS rates were 80.6%, 58.8% and 45%, respectively. The median recurrence-free survival (RFS) was 26 months (SE 3.18 [95% CI: 19.78–32.22]). The 1-, 3- and 5-year RFS rates were 67.4%, 44.2% and 35.7%, respectively.

The median OS in patients with recurrence (*n* = 226) was decreased compared to patients without recurrence (*n* = 161) (median OS 80 months vs. 49 months, *p* = 0.001).

### 3.2. Risk Factor for Recurrence

The univariate and multivariate analyses of prognostic factors for recurrence-free survival showed that tumor size >50 mm (HR 1.601 [CI95% 1.16–2.22] *p* = 0.05), satellite nodules (HR 1.467 [CI95% 1.01–2.13] *p* = 0.044), microvascular invasion (HR 1.62 [CI95% 1.17–2.26] *p* = 0.004), cirrhosis (HR 1.70 [CI95% 1.23–2.35] *p* = 0.001), bilobar location (HR 2.03 [CI95% 1.29–3.29] *p* = 0.002) and adjacent organ invasion (HR 4.97 [CI95% 1.99–12.41] *p* = 0.001) were independent risk factors for recurrence.

### 3.3. Patterns of Recurrence

Among 226 patients who developed recurrence, 104 (46%) had an early recurrence and 122 (54%) a late recurrence, with a cut-off for early recurrence set at 11 months, according to the “minimum-*p*-value” method (Appendix A). Among the late recurrence group, 18 patients recurred very late, beyond five years of monitoring. Recurrence consisted of a single nodule in 42% of cases. It was strictly intrahepatic in 73.5% of cases, with 16 type 1 in contact with the surgical margin, 36 type 2 in an adjacent segment, 46 type 3 in a distal segment and 68 type 4, multifocal. Recurrence was strictly extrahepatic in 10.2 % and mixed (intra + extra-hepatic) in 16.4%.

At recurrence, 120 patients (53.1%) were in the Milan criteria and 139 (61.5%) had an AFP score ≤2, eligible for liver transplantation. Child–Pugh’s recurrence score was A for the majority of patients (87.2%) and about half of the recurrences (52.2%) were classified as BCLC stage 0/A.

Curative-intent treatments for recurrent disease were performed in 101 patients (44.7%): 51, 37 and 31 patients underwent thermal ablation, repeat hepatectomy and liver transplants, respectively. Palliative-intent treatment was performed in 100 patients (44.2%), 50 of whom underwent TACE, 51 systemic therapies and 17 other therapies such as radiotherapy, SIRT and intra-arterial chemotherapy. Best supportive care was given to 25 patients (11.1%) because they were not eligible for specific therapies (Figure 1).

### 3.4. Predictive Factors of Survival after Recurrence

Median survival after recurrence was 26 months (SE: 3.28 [CI95%: 19.57–32.43]).

Male sex (*p* = 0.014), albumin level ≤ 35 g/L (*p* = 0.05), platelet count ≤ 100,000 (*p* = 0.026),pre-operative AFP > 200 (*p* = 0.004), microvascular invasion (*p* = 0.006), recurrence size > 50 mm (*p* = 0.019), extrahepatic recurrence(*p* = 0.043), Child–Pugh B or C score at recurrence (*p* < 0.0001), early time to recurrence and a not-curative treatment for recurrence (*p* < 0.0001) were independently correlated with decreased survival after recurrence according to the Cox model.

The results of the univariate and multivariate analysis of the variables significantly associated with SAR are shown in Table 2.

### 3.5. SAR According to the Treatment of Recurrence

The survival rate after recurrence at 1, 3 and 5 years was, respectively, 95%, 68% and 53.1% in patients who received curative treatment, compared to 62.9%, 23.4% and 12.7% in those with palliative treatments and to 8%, 4% and 0% in patients who were given the best supportive care (*p* < 0.0001) (Figure 2). Liver transplantation allowed the best survival benefit after recurrence (median SAR 158 months). However, our study did not report any significant difference in terms of SAR between thermal ablation and repeat hepatectomy at recurrence (median SAR 62 months vs. 42 months, *p* = 0.187).

### 3.6. Survival Analysis According to Time to Recurrence

#### 3.6.1. Early vs. Late Recurrence

Compared to late recurrence, early recurrence after initial surgery was a major prognostic factor leading to a decrease in SAR. Median and five-year SAR were 17 months (SE 1.70 [CI95% 13.67–20.33]) and 21.4% for early recurrence vs. 40 months (SE 5.38 [CI95% 29.46–50.55]) and 37% for late recurrence; *p* = 0.003.

Early recurrence was less often available for curative treatment at recurrence (*p* = 0.029) and it was more often associated with microvascular invasion of the initial tumor (*p* = 0.01), initial preoperative AFP > 200 ng/mL (*p* = 0.02), multinodular initial tumor (*p* = 0.017), satellite nodules (*p* = 0.05) and extrahepatic recurrence (*p* = 0.037) than late recurrence.

#### 3.6.2. Impact of Curative Treatments for Early Recurrence on Survival

For patients with early recurrence (*n* = 104), median SAR and five-year survival rate after curative treatment (*n* = 40) were 42 months (SE 14.01 IC 95% [14.53–69.47]) and 41.5% versus 10 months (SE 1.6 CI95% [6.74–13.26]) and 9.1 % in patients without curative treatment for early recurrence (*n* = 64).

The analysis of survival from initial surgery showed a decrease in overall survival in patients with early recurrence (*n* = 104; median OS 25 months; CI 95% [21.68–28.32]) compared to patients without recurrence (*n* = 161; median OS 55 months; CI 95% [34.5–75.50]) *p* = 0.003.

However, the findings of a stratified overall survival analysis of the recurrence treatment showed no difference in overall survival between the patients in the early recurrence group who could benefit from curative treatment for recurrence and those who never experienced recurrence (*p* = 0.768), even after excluding patients who died within 90 postoperative days (*p* = 0.593) (Figure 3).

Patients in the early recurrence group who could not be curatively treated for their recurrence had a significantly lower overall survival rate compared not only to patients who did not recur (*p* < 0.0001), but also to those with early recurrence who underwent curative treatment (*p* < 0.0001) (Figure 3).

#### 3.6.3. Impact of Curative Treatments for Late Recurrence on Survival

No difference in overall survival was reported between patients with late recurrence (*n* = 122) and those without recurrence, whatever the treatment of recurrence. However, among the patients with late recurrence, there was a significant difference in the SAR depending on the type of recurrence treatment: the median SAR was 80 months (IC 95% [47.81–112.19]) in curatively-treated patients versus 23 months (IC 95% [13.10–32.90]) and *p* < 0.0001 in those who could not receive a curative treatment.

No significant difference in the studied variables was noted between the patients with late recurrence and those who never recurred.

## 4. Discussion

Among the 387 patients of this single-center retrospective study, 226 (58.4%) recurred, despite a high R0 resection rate, long monitoring and an acceptable postoperative mortality rate of 5.4%, comparable to that of other international surgical series [9,10,11]. In our study, the risk factors for recurrence and the prognostic factors for SAR confirmed those found in other studies [9,20]. The interest and originality of our study were to assess the impact on survival of recurrence treatment stratified on time to recurrence.

Early recurrence is associated with poor prognosis, and it is associated with more aggressive tumor characteristics. Our study concurs with this finding that early recurrence has a pejorative prognostic factor for SAR (HR:1.50 IC95% [1.07–2.10] *p* = 0.019) and that it is more frequently associated with tumor aggressiveness features (AFP > 200, MVI, satellite nodules, multinodular tumor).

These results reinforce the hypothesis that early recurrence is associated with occult micro-metastases of the initial tumor. By contrast, we demonstrated that when early recurrence was available for curative treatment, not only was survival after recurrence good (median SAR 42 months, 1-, 3-, 5-year SAR: 92.5%, 55%, 41.5%, respectively), but the OS of these patients was also equivalent to that of patients who had not recurred (median OS *p* = 0.768). Although these findings suggest heterogeneity in this subgroup of patients, they did not allow us to identify precisely which patients with early recurrence were likely to develop a recurrence accessible to curative treatment. Further studies are therefore needed to answer this question.

It is classically acknowledged in the literature that late recurrence corresponds to a “de novo” tumor, associated with persistent underlying liver disease, active hepatitis or cirrhosis [33,34]. However, in their meta-analysis [35], Xu et al. also identified other risk factors for late recurrence (male gender, macro and microvascular invasion, tumor size > 5 cm, multiple tumors, satellite nodules). There are probably other criteria to refine/customize in order to better characterize this population.

We have shown that, whenever it is performed, curative treatment for recurrence yields comparable outcomes to those of non-recurring patients. In this context, the recurrence of HCC, appears as a chronic disease with several events occurring over time, and not as the palliative course of cancer with a poor prognosis.

Regarding the type of treatment for recurrence, our study reported that salvage transplantation allowed the best survival rates after recurrence (median SAR 158 months), even though it achieved lower results compared to “de principe” transplant for initial tumor [36,37,38]. However, the place of liver transplantations performed “de principe” after initial re-section in patients at high risk of recurrence is yet to be defined. Indeed, in the current context of graft shortage, current selection criteria prioritize patients at low risk of recurrence according to the Milan criteria or to the AFP score in France [39,40]. Transplantation still remains a valid treatment option for recurrence. We did not find any significant difference in SAR between thermal ablation treatment and repeat hepatectomy at recurrence (median SAR 62 months vs. 42 months, *p* = 0.187). Several retrospective series did not demonstrate any difference in survival between repeat hepatectomy (RH) and thermal ablation for small recurrences [24,41,42,43]. Likewise, in a large meta-analysis including 1020 patients, Erridge et al. [22] never showed a statistically significant difference in survival between the two treatments (thermal ablation versus repeat hepatectomy: HR 1.03, CI95% [0.68 to 1.55] *p* = 0⋅897). In a recent randomized controlled trial (RCT) including 240 patients with HCC recurrence [44], Xia et al. also did not report a difference in survival between repeat hepatectomy or thermal ablation; however, they showed that recurrence was more common, and that survival decreased for >3 cm recurrences treated with radiofrequency, suggesting repeat hepatectomy superiority for large-sized recurrences. These findings are comparable to the results of primary HCC treatments, so that it is legitimate to consider that the strategic approach to the treatment for recurrence should be the same as that for first tumors, according to the EASL management algorithm.

The main limitation of this study lies in its retrospective nature, which creates biases. It is also an observational and non-interventional study, insofar as the choice of treatment for recurrence is not randomized, but mainly influenced by HCC recurrence characteristics and natural history, which prompts to adopt a cautious approach when analyzing the results. Finally, the opening of this study to other centers would enhance the statistical power and significance of the results.

To improve patients’ prognoses, it is therefore essential to increase the possibility of curative treatment for recurrence. To this end, several approaches are to be considered. First, through the identification of patients at high risk of recurrence who may benefit from adjuvant therapy, which is however not currently available. There are also recent advances in immunotherapy in the management of advanced HCC [45] that open up interesting opportunities, with several phase III trials currently underway (IMBRAVE 050 [46], ESMERALD-2, CHECKMATE-9DX, KEYNOTE 937), and whose results are expected to be released soon. Other adjuvant treatment strategies, such as stereotactic body radiation therapy (SBRT) on the adjacent parenchyma of the tumor bed after resection, also hold promise to decrease recurrence. In a single-center RCT, Shi et al. showed an increase in disease-free survival (DFS) after SBRT in case of narrow margin <1 cm and associated microvascular invasion (MVI), comparable to that in patients without MVI [47]. In Eastern countries, post operative TACE, which has been studied in large cohorts of patients in adjuvant therapy after resection, also seemed to improve the prognosis in selected patients [48,49]. However, intra-arterial treatments are not common in our practice for adjuvant therapies and their place is yet to be defined.

Since adjuvant therapies are lacking, the challenge is to detect recurrence as early as possible, the best strategy consists of a close post-operative monitoring, with alternate MRI and chest CT scan associated with AFP dosage, according to our practice, which should be performed every three months for the first two years and then every six months. Because recurrence can sometimes occur very late (8% of recurrences in our study), this monitoring is to be maintained for life, especially in the case of persistent liver disease, which is the only known risk factor of late recurrence. Another major line of research relates to the modalities of post-therapeutic surveillance and detection of HCC, including the development and validation of new imaging and biology techniques. The development of Li-RADS imaging criteria [50,51,52] and the evaluation of hepato-biliary contrast agents [53] move in this direction, since they increase HCC detection accuracy and allow the standardization of practices.

The objective for the future is to arrive at a personalized medicine, in order to determine the best treatment, according to tumor aggressiveness factors, and ultimately improve the survival of HCC patients. A lot of translational research has been conducted for several years with the aim of developing personalized medicine through the genomic analysis of HCC patients. Each HCC shows high tumor genomic heterogeneity [54], with an average of 40 to 60 somatic mutations, some of which can predict recurrence and survival [55]. Hoshida et al. also reported molecular signatures from analyses of adjacent cirrhotic tissues which enable predicting of overall survival and late recurrence reflecting de novo carcinogenesis [56]. Finally, the principle of liquid biopsy [57] has been developed for several years, which consists in a non-invasive method to search for circulating biomarkers derived from the tumor, such as circulating tumor cells (CTC), circulating tumor nucleic acids without cells (cDNA, mRNA or cRNA), extracellular vesicles and tumor-educated platelets (TEPS). Several studies have demonstrated the potential of biomarkers for tumor response assessment, searching for tumor residue after treatment, detecting recurrence and assessing prognosis [58,59,60].

## 5. Conclusions

HCC recurrence after resection occurs in more than 50% of cases and early recurrence is a pejorative factor for survival after recurrence. However, regardless of the time of recurrence, curative treatments make it possible to achieve overall survival rates similar to those of non-recurring patients. Early detection of recurrence, aggressive and repeat treatments are therefore key to improve prognosis of patients with HCC.

## Figures and Tables

**Figure 1 cancers-15-00232-f001:**
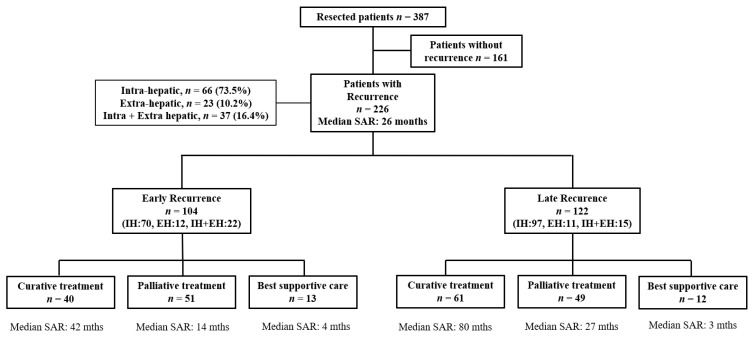
Flow chart of the study. SAR, Survival After Recurrence; IH, Intra-Hepatic; EH, Extra-Hepatic.

**Figure 2 cancers-15-00232-f002:**
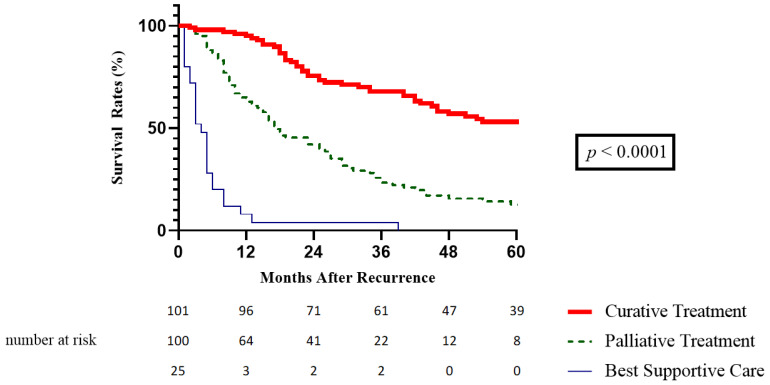
Survival after recurrence according to treatment of recurrence.

**Figure 3 cancers-15-00232-f003:**
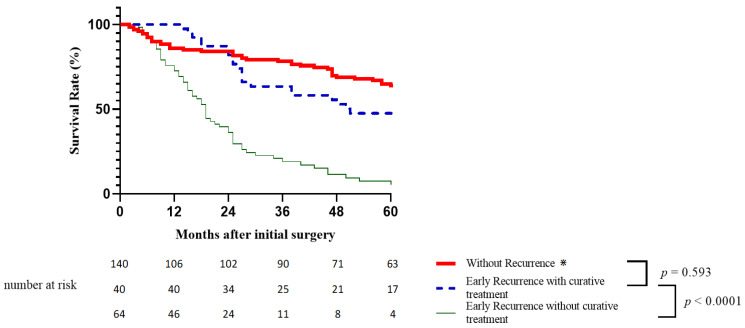
Overall survival according to early recurrence stratified by treatment. (⋇ after excluding patients who died within 90 postoperative days in this group).

**Table 1 cancers-15-00232-t001:** Comparisons of clinical characteristics between patients with or without recurrence.

Variables	Total *n* (%) *n* = 387 (100)	Without Recurrence *n* (%) *n* = 161 (41.6)	With Recurrence *n* (%) *n* = 226 (58.4)	*p*
Demographic Data				
Age > 70 years	117 (30.2)	45 (28)	72 (31,9)	0.409
Male sex	319 (82.4)	132 (82)	187 (82.7)	0.847
BMI > 30	68 (17.6)	30 (18.6)	38 (16.8)	0.601
F3-F4 Fibrosis	258 (66.6)	103 (63.9)	155 (68.6)	0.343
Child–Pugh Score A	346 (89.4)	143 (88.8)	203 (89.8)	0.621
HBV	17 (4.4)	5 (3.1)	12 (5.3)	0.297
HCV	97 (25.1)	32 (19.9)	65 (28.8)	0.047
Alcohol	131 (33.9)	57 (35.4)	74 (32.7)	0.586
NASH	21 (5.4)	11 (6.8)	10 (4.4)	0.303
Tumor Characteristics				
AFP > 200 ng/mL	49 (12.7)	20 (12.4)	29 (12.8)	0.803
Single nodule	307 (79.3)	133 (82.6)	174 (77)	0.103
Max Tumor size > 50 mm	140 (36.2)	53 (32.9)	87 (38.5)	0.261
Score AFP ≤ 2	234 (60.5)	98 (60.9)	136 (60.2)	0.879
Milan criteria in	232 (60)	96 (59.6)	136 (60.2)	0.913
Unilobar	353 (91.2)	152 (94.4)	201 (88.9)	0.010
Operative Data				
Pre-operative treatment (1)	86 (22.2)	39 (24.2)	47 (17.7)	0.500
Major hepatectomy	124 (32)	54 (33.5)	70 (31)	0.594
Anatomical resection	254 (65.9)	110 (68.3)	144 (63.7)	0.290
Blood transfusion	123 (31.8)	53 (32.9)	70 (31)	0.685
Laparoscopic approach	56 (14.5)	23 (14.3)	33 (14.6)	0.914
CCI > 26.2	131 (33.9)	61 (37.9)	70 (31)	0.191
Histological Data				
Satellite nodule	79 (20.4)	27 (16.8)	52 (23)	0.133
Poor differentiation	47 (12.1)	16 (9.9)	31 (13.7)	0.257
Microvascular invasion	129 (33.3)	44 (27.3)	85 (37.6)	0.028
R1 resection	25 (6.5)	5 (3.1)	20 (8.8)	0.022
Capsule	188 (48.6)	79 (49.1)	109 (48.2)	0.958
Adjacent organs invasion	8 (2.1)	0 (0)	8 (3.5)	0.023

(1): Including thermal ablation, trans-arterial chemoembolization, selective internal radiation therapy, systemic therapies. Abbreviations: BMI: body mass index; HBV: hepatitis B virus; HCV: hepatitis C virus; NASH: non-alcoholic steatohepatitis; AFP: alpha-fetoprotein; CCI: comprehensive complication index; R1 resection: when the surgical margin was positive or narrow margin < 1 mm.

**Table 2 cancers-15-00232-t002:** Univariate and multivariate analyses to predict survival after recurrence.

Variables	Univariate Analysis			Multivariate Analysis		
	Survival, Median (SE), Months	95% CI	*p*	HR	95% CI	*p*
	Variables at Primary Surgery					
Sex			0.095			
Female	35 (5.3)	24.7–45.3				
Male	23 (3.1)	16.9–29.2		1.92	1.14–3.24	0.014
Albumin (g/L)			0.012			
>35	29 (3.6)	22.0–36.0				0.050
≤35	18 (5.5)	7.2–28.8		1.74	1.00–3.04	
Platelet Count			0.037			
>100,000	29 (5.5)	18.2–39.8				
≤100,000	23 (3.5)	16.2–29.8		1.76	1.07–2.91	0.026
AFP (ng/mL)			0.020			
>200	29 (4.3)	20.56–37.44				
≤200	16 (5.4)	5.45–26.55		2.16	1.27–3.52	0.004
Hepatectomy			0.03			
Minor	36 (5.9)	24.45–47.55				
Major	16 (3.6)	8.99–23.01		1.38	0.87–2.20	0.174
Per operative Transfusion			0.043			
No	29 (5.3)	18.68–39.32				
Yes	18 (2.4)	13.22–22.78		1.15	0.77–1.72	0.503
Primary Tumor Size			0.001			
≤50 mm	40 (6.6)	27.06–52.94				
>50 mm	17 (3.5)	10.17–23.83		1.61	1.05–2.46	0.03
Microvascular Invasion			< 0.0001			
No	40 (3.8)	32.63–47.37				
Yes	17 (2.1)	12.83–21.17		1.72	1.17–2.53	0.006
Macrovascular Invasion						
No	29 (3.7)	21.73–36.27				
Yes	11 (5.3)	0.61–21.39	0.009	0.84	0.43–1.64	0.608
Complete Resection						
R0	27 (3.4)	20.37–33.63				
R1	9 (3.7)	1.70–16.30	0.024	0.61	0.28–1.34	0.221
Adjacent Organ Invasion						
No	27 (3.4)	20.39–33.62				
Yes	9 (3.8)	1.61–16.39	0.072	1.52	0.44–5.30	0.511
	Variables at Recurrence					
Number						
single	44 (7.4)	29.56–58.44				
multinodular	19 (2.8)	13.47–24.53	<0.0001	1.12	0.78–1.61	0.526
Size						
≤50 mm	34 (4.8)	24.51–43.49				
>50 mm	7 (1.9)	3.33–10.67	<0.0001	2.15	1.13–4.10	0.019
Location of Recurrence						
Intra-hepatically (IH)	40 (5.6)	29.07–50.93				
Extra-hepatically ± IH	14 (3.9)	6.41–21.60	<0.0001	1.53	1.01–2.32	0.043
Timing of Recurrence						
Late	40 (5.4)	29.46–50.55				
Early	17 (1.7)	13.67–20.33	0.003	1.50	1.07–2.09	0.019
Child–Pugh Score at Recurrence						
A	32 (4.3)	23.61–40.39				
B or C	5 (2.5)	0.10–9.90	<0.0001	4.47	2.71–7.38	<0.0001
Treatment at Recurrence						
Curative	61 (13.3)	34.87–87.13				
Not curative	14 (2.1)	9.92–18.08	<0.0001	2.98	1.99–4.46	<0.0001

Abbreviations: SE: standard error, CI: confidence intervals, HR: hazard ratio; AFP: alpha-fetoprotein.

## Data Availability

The data presented in this study are available on request from the corresponding author.

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
