# Peer review of "Prolonged Survival after Recurrence in HCC Resected Patients Using Repeated Curative Therapies: Never Give Up!"

_cancers, 2022, doi:10.3390/cancers15010232_

Round 1
Reviewer 1 Report
The authors persist aggressive treatment can make good outcomes in HCC treatment, however there are several comments regarding this article.
1. There must be reasons why patients could not receive aggressive treatments, but it was not analyzed in this article.
2. The authors suggest early detection is important, however there is no result that supports early detection is important for outcomes.
3. The authors suggest that early recurrence with curative treatment shows same outcome compared to no recurrence, but the figure 3. Suggest that if there is enough follow up period, no recurrence might show best outcomes.
4. English corrections should be considered. Eg.) There is no CHILD score, Child-Pugh score is the correct term.
5. Regarding these comments, I think the authors cannot suggest aggressive treatment cause good outcomes I suggest that this article should focus on natural history of HCC recurred patients
Reviewer 2 Report
This is a very nice study on long-term follow-up after liver resection for HCC, with a particcular focus on survival after recurrence. The final conclusion of the authors is that in any case of HCC relapse options of curative intent treatments should be evaluated, since these may significantly improve survival. The manuscript is well written and the data are very interesting and merit to be published.
I have only a few issues that should be addressed by the authors prior acceptance.
1. The authors should more detailed describe surgical procedures of primary resection, specifically definitions and numbers of major hepatectomy (hemihepatectomy, trisectorectomy) and minor reection (anatomic, non-anatomoc, mono/bisegmentectomy et).
2. The authors should include differential criteria for curative intent therapy between re-resection, slavage LT and RFA; in this context they should also discuss on waiting time for liver transplant and risk of drop-out. Fo example, how many patients were waiting for slavage LT and had to be switche to palliative treatments due to tumor progress.
3. It could be interesting which diagnostic device was most successful for early detection of HCC relapse; AFP versus CT/MRI?
4. Minor point: SAR should be explained in teh abstract section.
Reviewer 3 Report
Well written, highlighting an important concern in patients with HCC who undergo resection. no major concerns regarding the concept, methodology, results or discussion. Not an entirely new concept but interesting study that highlights the fact that inspite of high rates of recurrence post resection, even in patients with early recurrence which essentially is poor prognosticator, early diagnosis and agrressive treatments can still imporve outcomes.
Reviewer 4 Report
1. In section Introduction, the reference 2 (Mathers and Loncar, 2006) discussed mainly about population health trends and health policies. The evidence of hepatocellular carcinoma is minimal in this reference. I think this citation is not feasible.
2. In 2.2 Clinical Variables, section Materials and Methods, authors mentioned about the type of hepatectomy (major or minor). In previous articles, different definitions of major liver resection were declared. Please define the major and minor hepatectomy in your manuscript.
3. In 2.2 Clinical Variables, section Materials and Methods, one variable is degree of associated liver fibrosis. In table 1, authors highlight F3-F4 Fibrosis. According to Chapter 22, Liver, AJCC Cancer Staging Manual 8th edition, fibrosis scores include Batts-Ludwig, Ishak, and METAVIR systems. Please declare what score system is used in section Materials and Methods.
4. In table 1, one factor is Score AFP ≦2. According to reference 5 (Duvoux et al., 2012), does Score AFP means the simplified, User-Friendly Version of the AFP model (table 2 in Ref 5)?
5. The cut-off values of tumor size and AFP differ between table 1 and 2. In table 1, cut-off values of tumor size and AFP are 30mm and 100 ng/ml. In table 2, cut-off values of tumor size and AFP are 50mm and 200 ng/ml. Is there any specific perspective? I suggest cut-off value should be unified in a manuscript.
6. In table 2, less variables are documented at recurrence. By time progressing, biologic factors (e.g. albumin) at recurrence may differ from those at primary surgery. Do authors ever consider these also influencing the prognosis?
7. In figure 3, patients with early recurrence are categorized in with/without curative treatment and overall survival rates are significantly different. I suggest authors discussing about the reason of treatment strategy in section Discussion. Why do certain patients receive curative treatment and others not? Authors’ perspective may be valuable for clinical physicians.
Reviewer 5 Report
The experience presented by the authors is important due to the number of patients and is well collected since the study includes a significant number of variables. The article is well written and without being an expert in statistics I consider that the methodology is adequate.
The most relevant conclusion is that early recurrence implies a worse prognosis. However, if the treatment for recurrence is curative (resection, ablation or liver transplantation), it is possible to achieve a survival similar to that of patients without recurrence. This is a very interesting conclusion, since it justifies the implementation of an intense follow-up program after surgery and the indication of aggressive treatment policies.
On the other hand, and putting myself in the place of future reader, I consider that the authors should clarify or deepen a series of aspects.
The first aspect is somewhat philosophical. It seems evident that the possibility of applying a curative treatment of an early recurrence (single tumor, accessible to a minor resection or ablation) is probably a marker of less biological aggressiveness of the disease. Although this concept does not change the strategy suggested by the authors, it can question a part of the title (“never give up”). It is the way in which the recurrence is presented that establishes whether we should give up.
Among demographic data, HCV infection was the only significant variable. Hepatitis C virus infection was more frequent among patients with recurrence. However, the differences were close to not being significant. Were these patients treated for their viral infection before or after the intervention?
In view of the results, would the authors indicate ablation as the treatment of choice for single recurrences with nodules smaller than 3 cm?
In the discussion, the authors could comment on their opinion about "ab initio" liver transplantation in patients with a high risk of recurrence after resection.
What do the authors think about the follow-up of patients with healthy livers? Should it also be permanent?
There may be an error in the recurrence-free survival data found on line 175 on page 5. The 5-year recurrence-free survival should not be higher than that seen in the first year. Maybe it's recurrence data.
Round 2
Reviewer 4 Report
I have no other question.